# Traditional Food Products—Between Place Marketing, Economic Importance and Sustainable Development

**Magdalena Florek** \* and **Jakub Gazda**

The Department of Commerce and Marketing, Poznań University of Economics and Business,
Al. Niepodległości 10, 61-875 Poznań, Poland; jakub.gazda@ue.poznan.pl
\* Correspondence: magdalena.florek@ue.poznan.pl; Tel.: +48-61-856-94-24

**Abstract:** The aim of this paper is to link the economic and marketing perspectives by means of the quantitative method to answer the question of whether traditional food products may play an important role in sustainable region development ensuring economic viability, and how (if at all) marketing should support this process. Due to the lack of unambiguous theoretical findings—the theory has not established a model of the impact of regional products on the food sector in the region and, further on, the economy—an atheoretical approach should be applied, which without priori findings (without favouring individual variable) will lead to an assessment of the above-described impact. Using the Bayesian approach, the authors have measured the above relations with respect to the case of Poland. The basis of the study is provided by a database developed by the authors for 16 Polish NUTS2 regions where lists of official traditional food products were compiled. Using the chosen method, a group of explanatory variables has been proposed (among them, a number of regional products) as factors potentially responsible for the dependent variable (value of agriculture production in the NUTS2 regions). The results indicate that the number of traditional food products does not determine the value of agricultural production on the level of the NUTS2 regions in Poland. This value is determined by commodity production rather than the factors of the so-called sustainable agriculture.

**Keywords:** traditional food products; List of Traditional Products; Bayesian model averaging; place marketing

## 1. Introduction

According to an analysis following the Food and Agriculture Organization of the United Nations (FAO) and carried out by Vandecandelaere et al. [1], the methodology of the virtuous circle of origin-linked quality and geographical indications (GIs) can be applied to support sustainable development and sustainable food systems. Having the potential to promote economic development and food security, they can even be of assistance in achieving the Sustainable Development Goals (SDGs). Food is the common denominator linking all the 17 United Nation SDGs, given the interconnected economic, social and environmental dimensions of food systems. Tellström et al. [2] acknowledge that traditional foods have been used as vehicles to promote regional economic growth and food is a key component in communicating culture. In a report "Food in an Urbanised World" [3], the authors argue that while food system challenges have many global dimensions, a "city region food system" approach is a reasonable way of addressing the challenges attributed to specific places, in terms of causes, impacts and governance. As globalization entails a process of glocalization [4] where locally available resources—geographical and cultural—have the potential to build unique value to consumers [5], the role of branding is growing, as local food is considered in two dimensions: as a material product and as a specific place's intangible heritage (Yurtseven and Kaya 2011, cited in [6]). Therefore, traditional regional food can bring competitive advantage, and as such, it is important to recognise the actual influence of the place (region) of origin on the value of production of

agricultural products and, consequently, on a region's development. It reflects the validity of marketing endeavours underpinning protection and promotion of a region's products. In this approach, ensuring economic viability is a factor of key importance, but empirical evidence of the benefits of GIs is scant, especially in countries where the GI procedures have been recently introduced [7].

This paper takes a broader perspective of the issue in an attempt to evaluate the impact of the number of local traditional products in regions (as included in the List of Traditional Products) on the general value of food production (value of agriculture production) in regions.

The aim of this paper is to answer the question of whether traditional food products may play an important role in sustainable region development ensuring economic viability, and how (if at all) marketing should support this process. Due to the lack of unambiguous theoretical findings, as the theory has not established a model of the impact of regional products on the food sector in the region and, further on, the economy, an atheoretical approach was applied by the authors, which without priori findings (without favouring individual variable) will lead to an assessment of the above-described impact. The paper, therefore, fills the methodological gap in the evaluation of the impact of particular variables on food production in a region. The following hypothesis has, therefore, been formulated: on the NUTS2 (Nomenclature des unités territoriales statistiques) level in Poland, the number of traditional regional products determines the value of agriculture production. Using the Bayesian approach, the authors have measured the above relations with respect to the case of Poland. The basis of the study is provided by a database developed by the authors for 16 Polish NUTS2 regions where lists of official traditional food products were compiled. The List of Traditional Products was compiled by the Ministry of Agriculture and Rural Development in Poland. Official traditional products are registered in all the 16 regions (NUTS2) in Poland (the list differs, however, from the EU schemes known as the PDO-protected designation of origin, the PGI-protected geographical indication and the TSG-traditional specialty guaranteed that promote and protect names of quality agricultural products and foodstuffs). The intention behind the List of Traditional Products is to collect and disseminate information related to the manufacturing of traditional products. Applying the chosen method, a group of explanatory variables has been proposed (among them a number of regional products and variables related to sustainable development) as factors potentially responsible for the dependent variable (value of agriculture production).

The study is limited by the geographical scope (one country) and a purely quantitative approach.

First, the background for the research is presented, followed by the methodology applied. Next, the achieved results are discussed, and finally, conclusions in the place marketing context are briefly offered.

### 1.1. The Place of Origin Effect, Place Marketing and Sustainability

Throughout the world, consumers, producers and public authorities are showing a growing interest in food and agricultural products linked to their place of origin (Barham and Allaire 2011, cited in [7]). The country of origin has been extensively analysed, with more than 800 academic publications related to business–branding–places effects [8]. The majority of academic research focuses on analysing the relations between consumers' associations with a place of origin and their willingness to buy products or brands based on this cue. As such, the consumer behavioural perspective is the most popular in literature on the subject.

Consumers are willing to perceive and evaluate the quality of a product based on its place of origin, transferring opinions about the place and the related attitudes onto specific goods or services. This phenomenon is particularly remarkable in countries (when a country's image determines the perceived quality of a product category and stimulates its sales) and is known as the country-of-origin effect (COO). However, lately, a broader concept, namely, the place-of-origin effect (POO), seems to be more relevant, as the effect is

valid also for regions, small distinctive areas and even cities specializing in production of certain goods.

Indeed, studies of the effect of the COO have focused on numerous durable and non-durable consumer goods. Research results have invariably confirmed the fact that consumers tend to use information on the country of origin as a quality identifier. This impacts consumers' attitudes even when they have an opportunity to see, touch or taste very similar products [9]. Notably, consumers make use of the country-of-origin effect not only as information as such but also as a source of information about other products. Verlegh and Steenkamp (1999), as cited in Tell [2]), noted that the definition of the COO should exceed the cognitive cue of product quality. The COO also encompasses emotions, identity and personal memories that transform the COO into an image attribute.

Verlegh [10] suggests that the geographic components of a place image, such as the climate and the natural landscape, influence consumers' beliefs, especially about the food products. Tellström et al. [2] acknowledge that, associated with a place, food will transmit the inhabitants' heritage and identity. Moreover, the branding of local foods emphasizes their symbolic and cultural meaning. For expatriates, food may be a source of national pride, while for former visitors to a region it might elicit memories of past experiences. What is more, local agricultural production contributes to re-introducing and maintaining the local identity and culture as well as reinforcing community pride and re-establishing the local identity and culture [11]. A report drawn by the World Tourism Organisation (2012) stressed the importance of food as immaterial cultural heritage that can enhance the reputation of places worldwide and differentiate locations. Using local food and drinks allows regions "to incorporate cultural distinctiveness within economic development" (Haven-Tang and Jones 2005:71, cited in [6]). An opportunity avails itself, especially for developing regions, to capitalize on consumers' perceptions of them as more old-fashioned and, thus, more natural and wholesome. According to Anholt [12] (p. 147) "consumers are also becoming more demanding about the authenticity of the ethnic food they buy rather than products with a manufactured spin".

The possibilities offered by the POO for place development have been so far discussed in several contexts, e.g., how the image of a place impacts the image of the products originated in that place (see [13]), how the image of the products and brands impact the place image [14], the impact on the sales of regional products or brands as a result of the so-called buy-national campaigns (e.g., [15–17]).

The mutual influence (transfer) of a place image (a region's image in this case) on the specific products manufactured in situ is sometimes an automatic, self-activating process. More and more frequently, it has been intentionally used in order to achieve a region's superior or particular goals.

Regional food culture has, therefore, become a tool not only of promoting the economic and rural growth in regions suffering recession [18–20] but also of branding places. A review of 170 published studies on city branding [21] shows that elements of food, beverages and the HoReCa channel are used frequently as elements of city branding.

According to Richards (2015, cited in [6]), food may be an essential element in branding places, since it involves and connects many aspects of experience. This experience can refer to many "dimensions": products (food and beverages), practices (eating and meals), the art and customs of preparing and eating (gastronomy), sensory elements (taste, smell, touch, visual), the origin (organic food, ethical cuisine, locally produced food, etc.), preparation (ways of cooking), serving (fast food, slow food, street food, etc.) and the context in which food is served and consumed (restaurants, bars, markets, food quarters, streets, etc.). The Organisation for Economic Co-operation and Development [22] revealed that food plays an important role in the development of tourism services, since it often represents 30% or more of tourist spending and the money is spent directly on local business. Tourists' consumption of local food supports the local economy [23] and tourist spending on locally produced goods may stimulate the local economy to maintain and/or reinvigorate the

local primary production and processing sectors (Du Rand et al. 2003, Boyne et al. 2003, cited in [6]).

A study carried out by Berg and Sevon [24] suggests three main categories of arguments in favour of being associated with attractive food products and meals: (i) support of the food industry, (ii) protection and reinforcement of the identities of places and (iii) transformation of the place. What is more, UNESCO has analysed the recognition of areas where humans have shaped the landscape in harmony with nature to produce good like food and drinks [25].

### 1.2. Geographical Indicators

International bodies, such as the FAO of the United Nations, have analysed the relationship between products and territories under the denominations of geographical indications [1,7].

Geographical indications (GIs) are defined by the EU as "a sign used on products that have a specific geographical origin and possess qualities or a reputation that are due to that origin. In order to function as a GI, a sign must identify a product as originating in a given place. In addition, the qualities, characteristics or reputation of the product should be essentially due to the place of origin. Since the qualities depend on the geographical place of production, there is a clear link between the product and its original place of production" [26].

According to Giovannucci et al. (2009, cited in [7]), there are more than 10,000 geographical indications in the world, mostly in the agricultural and food sectors, with an estimated trade value of more than USD 50 billion. Many are well-known worldwide, such as Darjeeling tea or Parmigiano-Reggiano cheese, but many more are famous in their domestic markets, while some are anticipating a boost to their reputation from the GI registration [7]. The "taste of a place" [27] implies that geographic conditions contribute to food's characteristics and qualities [6]. Although origin-linked products may be referred to in a variety of ways (terroir products, traditional products, regional foods, genuine products, etc.), they all attach their value to their origin (Barham and Allaire 2011, cited in [7]).

The economic and political use of food culture is emphasised by the EU, with some regional food cultures used as a concept in political projects since the 1990s (Delanty 1998, Ilbery and Kneafsey 2000, cited in [2]). This has been expressed through protective legislation on food product origin (CEC 1992) and in programmes for rural development such as artisan food production (Tregear 1998, cited in [2]).

The link to the origin results from a combination of local resources, i.e., natural resources (species or breeds, soil and climate conditions, landscape, micro-environment, etc.) and human and cultural resources (local know-how, history, traditions) [7]. As D'Amico [28] (p. 794) summarized "the historical dimension concerns cognitive content, which is knowledge and know-how consolidated over time", it is, therefore, related to customs of how typical products are produced, processed and consumed, making them part of both a local heritage and a place's history [6]. As such, the GIs recognition enables consumers to trust and distinguish quality products while also helping producers to market their products better [29].

As acknowledged by [7], GIs can be also important drivers of rural transformation leading to more sustainable development (FAO 2016; Durand and Fournier, 2015, cited in [7]). According to them, this is "first because economic sustainability is an important step towards environmental and social sustainability (positive environmental and social impacts of GIs cannot be supported if producers have to abandon their practices to be more competitive). A second reason is that the specifications can directly influence environmental sustainability, depending on the requirements that are considered (local species or breed, specific agricultural practices etc.)" (p. 33).

The EU quality policy aims at protecting the names of specific products to promote their unique characteristics, resulting from their geographical origin as well as traditional

know-how. Product names can be granted "geographical indication" if they have a specific link to the place where they are made.

The European Union provides legislation schemes for protecting products associated with specific places under the following labels: (1) protected designation of origin (PDO), (2) protected geographical indication (PGI), (3) traditional specialities guaranteed (TSG). In addition, there is a geographical indication of spirits and aromatised wines.

Products registered under these schemes may be marked with a logo with the purpose of certification and identification. The schemes are based on a legal framework provided by Regulation no. 1151/2012 of the European Parliament and of the Council of 21 November 2012 on quality schemes for agricultural products and foodstuffs. This regulation (enforced within the EU and gradually expanded on an international level by means of bilateral agreements between the EU and non-EU countries) ensures that only products genuinely originating in a specific region are allowed to be identified as such in commerce. The purpose of the law is to protect the reputation of regional foods, promote rural and agricultural activity, help producers obtain premium prices for their authentic products and eliminate unfair competition and misleading of consumers by non-genuine products, which may be of inferior quality or of a different flavour.

Conclusions from the analysis of the nine cases presented by [7] confirm (1) specific quality formally indicated in specifications or a code of practice, (2) capacity for collective action and good governance, (3) an effective marketing strategy and (4) a legal/institutional framework as the factors of successful GIs implementation. Specifically, three main marketing strategies have been identified as the key success factors [7]:

1. GI branding: the capacity to reach agreement with downstream actors is an essential element in enjoying economic impacts. As the case of Colombian coffee shows, branding strengthens the visibility of a GI product and promotes the correct use of its registered name in the point of sale.
2. Targeting of niche markets: the marketing strategy is driven by the type of GI approach (offensive or defensive) and the marketing channel (niche or mass). The best economic impacts are seen when a GI organization adopts a strategy of managing the volume of supply to prevent bringing down of prices resulting from significant increases in volume, with production exceeding demand.
3. Gaining access to new markets in times of change: developing or conquering new (niche) export markets can help to avoid the effects of a national-level crisis. This was the case of Manchego cheese, which avoided the full impact of a domestic economic crisis by expanding exports to the United States.

As Vandecandelaere et al. [7] concluded, official GI recognition and registration act as incentives, both to value chain stakeholders (producers and downstream players) to create and perceive values and to public authorities to generate and enhance public goods.

However, due to the large number of the determinants affecting the value of production, identification of the extent and range of impact and the plausibility of creating demand for regional goods is a complicated issue. At the same time, identification of these relations may become an important premise for making various types of decisions in a region (e.g., supporting promotion) and in the local companies.

## 2. Materials and Methods

### 2.1. The System of Regional Products Registration in Poland

Some countries, including Poland, have introduced and promoted their own products' signage, emphasizing the products' association with a country or region. The underlying goal is to both protect traditional products and to propagate them, including the relations between food products and the place of their production, taking advantage of the image transfer.

In Poland, the Ministry of Agriculture and Rural Development is responsible for operating the system of registering products of a specified geographic origin and a specific, traditional value as stipulated in the European Union regulations.

According to the legal act of 17 December 2004 on the registration and protection of names of agricultural products and foodstuffs, the Ministry of Agriculture and Rural Development is in charge of collecting, evaluating and forwarding applications for registration of names of origin, geographic signage and guaranteed traditional specialties to the European Commission. On top of the regulations behind registering names on the EU level, the act also provides a List of Traditional Products. The list includes products whose quality or unique features and characteristics result from applying traditional production methods, are elements of a region's cultural heritage and are a part of the local community's identity. The requirement is that the traditional production methods have been applied for at least 25 years. The age of the method needs to be corroborated by means of a book, a photograph or a recording. When these tend to be obscure, the applicant may carry out ethnographic research, e.g., conversations with the elderly, witnesses to consumption of a given specialty [30] (p. 11). A product whose manufacturer applies for inclusion thereof in the List of Traditional Products should be an element of the local community's identity and be a part of a region's cultural heritage. The producers do not acquire any rights, nor do they qualify for protection or promotion of the enlisted products as their own, because it is the product that is protected, not the manufacturer. A province marshal is in charge of verifying applications to include a product in the List of Traditional Products. Before the evaluation, the marshal addresses a chamber of commerce whose members are manufacturers of regional and traditional products to express their opinions. Later on, the application is forwarded to the Ministry of Agriculture and Rural Development. The Minister is in charge of compiling and updating the List of Traditional Products. The list is published in the Minister's Official Journal and on the website of the Ministry of Agriculture and Rural Development (regularly).

The list has been created for meat, dairy, fish, vegetable and fruit products, pastries, fats, oils, honey, beverages, meals, etc. The first product included in the list (July 2005) was pierekaczewnik, a pie dish from the Podlasie region enjoyed by the local Tartars. In 2009, pierekaczewnik was registered in the EU as a Guaranteed Traditional Specialty so that the concept of the List of Traditional Products as a "preface" to the much more complicated EU registration has proven true [31].

To sum up, the goal of the List of Traditional Products is to propagate information about products manufactured by traditional methods with historical background. The List of Traditional Products includes 1993 products (as of 16 October 2020) from all Polish regions [32]. In order to maintain a homogenous timeframe of the research, the applied research methodology requires reference to the number of traditional products at the end of the research time, i.e., 21 December 2018 (see Table 1).

### 2.2. The Research Method

The goal of the research is to check the factors determining the value of agricultural production in the NUTS2 regions in Poland, i.e., to check if the number of traditional protected regional products determines the value of agricultural production on the level of the NUTS2 regions. Another issue is how, in the probability ranking, the variables characteristic of sustainable agriculture (for example, "Ecological certified farms-share of the area of cultivated land in total cultivated land", see Table 2) act among other variables.

This way, we can indirectly define if the marketing activities accompanying regional products, including the signage system and promotion of traditional products, affect a broader economic context, and if they are important drivers of transformation, leading to more sustainable economic development.

**Table 1.** Number of traditional products of the Ministry of Agriculture and Rural Development as of the end of 2018.

| Region (Voivodship, NUTS2) | Number of Traditional Products |
|---|---|
| Dolnośląskie | 49 |
| Kujawsko-pomorskie | 83 |
| Lubelskie | 208 |
| Lubuskie | 78 |
| Łódzkie | 141 |
| Małopolskie | 218 |
| Mazowieckie | 132 |
| Opolskie | 62 |
| Podkarpackie | 229 |
| Podlaskie | 69 |
| Pomorskie | 178 |
| Śląskie | 145 |
| Świętokrzyskie | 92 |
| Warmińsko-Mazurskie | 34 |
| Wielkopolskie | 93 |
| Zachodniopomorskie | 53 |

Source: Ministry of Agriculture and Rural Development in Poland.

**Table 2.** List of variables chosen for the research process.

| Symbol | Variable |
|---|---|
| Y | Value of agricultural production per 1ha in the NUTS2 regions in 2018 |
| X1 | Number of regional products in the NUTS2 regions as of end of 2018 |
| X2 | Number of regional products in the NUTS2 regions (av. 2010–2018) |
| X3 | Value of agricultural production per 1ha in the NUTS2 regions in 2010 |
| X4 | Ecological certified farms-share of the area of cultivated land in total cultivated land (av. 2010–2018) |
| X5 | Inner investments in biotechnological activity in companies (av. 2010–2018) |
| X6 | GDP per capita value in the NUTS2 regions in 2010 |
| X7 | Average population density (av. 2010–2018) |
| X8 | Average monthly per capita expenditures on food and soft drinks (av. 2010–2018) |
| X9 | Per capita capital expenditures (av. 2010–2018) |
| X10 | Average share of commodity production in agricultural production (av. 2010–2018) |

Source: own compilation.

The literature on economic growth, e.g., [33–35], encompasses a range of studies referring to various factors and groups of factors responsible for the processes of economic growth. These studies provide the foundation for the considerations presented below. There is consent in literature on the subject that the methods developed on the basis of Bayesian econometrics are generally applicable in an analysis of a complex economic phenomenon such as the determination of the sources of economic growth but also might be applicable as a research methodology to new fields without an established theory [36]. In relation to the above, the Bayesian method was used in the research described below.

Bayesian inference, along with the multi-channel Markov chain Monte Carlo (MC3) algorithm, allows us to select the most likely combination of independent variables (i.e., factors responsible for the processes of economic growth) from a very large set of variables. It also contributes to a calculation of the explanatory power of all the interesting models and their ranking from the most to the least likely one, and the averaging of posterior estimations (including the mean and variance), weighted with the posterior probability of the models.

Why is it necessary to apply the Bayesian pooling approach and numerical techniques for a simple regression model if the estimation results can be obtained analytically, provided that prior distributions are selected accurately [37]? The answer is quite simple. When the number of independent variables in a regression model is very large, it is extremely time-consuming, or virtually impossible, to compute all the possible combinations of these variables. Moreover, it frequently turns out that the model with the greatest explanatory power has slight posterior probability. When we focus on this model exclusively, we actually ignore a stream of additional information provided by other models whose total posterior probability can be very high. Madigan and Raftery [38] additionally indicated that the popular methods of choosing variables applied in the classical approach can lead to different selections of independent variables and, thus, to different conclusions.

Let us consider, for example, a regression model with three potential independent variables $X_1$, $X_2$, $X_3$. In this case, we obtain $L = 2^3 = 8$ linear combinations of independent variables. They can be listed as follows:

$$M_1 : y = a_0 + e, \quad M_2 : y = a_0 + a_1 X_1 + e, \quad M_3 : y = a_0 + a_2 X_2 + e,$$

$$M_4 : y = a_0 + a_3 X_3 + e,$$

$$M_5 : y = a_0 + a_1 X_1 + a_2 X_2 + e, \quad M_6 : y = a_0 + a_1 X_1 + a_3 X_3 + e,$$

$$M_7 : y = a_0 + a_2 X_2 + a_3 X_3 + e, \quad M_8 : y = a_0 + a_1 X_1 + a_2 X_2 + a_3 X_3 + e$$

Let us assume that a random element has a normal distribution. Given the conjugate prior distributions, the estimation of the parameters in all the above-listed models can be made analytically, without a need of numerical methods (the concept of conjugate distributions says that if the prior distribution of a parameter in question belongs to a given family of distributions, then its posterior distribution also belongs to the same family for any size of an $n$ sample and any number of observations. An ideal family of distributions is one that allows us to obtain an easy point estimation of a parameter and one flexible enough to easily express the initial information). The adopted assumptions also allow for an analytical calculation of the explanatory power of competitive models, and to determine the most probable posterior one.

In the classical approach, a typical procedure for the regression model construction involves an estimation of the model's parameters (e.g., using the classical method of least squares—CMLS) followed by rejection of insignificant variables so that a single accurate model is designed. In this approach, the uncertainty related to the explanatory value of a model is ignored, which means that estimating its probability is impossible. The difference between Bayesian inference and the classical approach stems from the fact that, among other things, the former takes into account the uncertainty related to the selection of a model by means of calculating its posterior probability. Let us assume that model five ($M_5$) obtained the highest explanatory power with posterior probability amounting to 0.3. All the remaining models had a smaller explanatory power, but their total likelihood amounted to 70%. If we analyse only one model, we will ignore an abundance of additional information included in the remaining models. That is why it is sometimes necessary to apply the Bayesian pooling approach, which involves, among other things, the averaging of parameter estimations and their posterior distributions weighted with the posterior probabilities of individual specifications. Conclusions can be drawn on all the interesting values not only on the basis of a single model but all the models, consistent with their explanatory power.

When a set of potential independent variables comprises 30 elements, the number of the possible combinations increases to as many as $L = 2^{30} = 1,073,741,824$. Assuming that computing each combination takes only one second, it would take as long as 34 years to calculate all of them. Therefore, a more efficient algorithm is required to compute combinations, in order to focus on the most likely variants and ignore those with negligible posterior probability. That is the purpose of the $MC^3$ algorithm developed by Madigan,

York and Allard [39]. The elements of Bayesian inference, the estimation procedure of the regression model along with the MC$^3$ algorithm have been broadly presented by Błażejowski et al. [40] and Gazda and Puziak [41]. The research procedure described in detail in the above-mentioned works has been applied in order to identify the sources of economic growth in EU regions.

On the theoretical level, the above presented procedure has been described in an article by Błażejowski et al. [35]; a detailed description of the tools is contained in a publication by Błażejowski and Kwiatkowski [42].

Regional agricultural production may potentially be influenced by an enormous number of factors. A simple attempt at listing them poses problems with classifying and indicating the division criteria. The selection has, therefore, been made on the basis of earlier empirical studies carried out by the authors [43].

The authors of the study have assumed that the influence of the value of agricultural production related to regional products might not be very significant; this study is only a beginning of an investigation into how important (if at all) regional products (and their branding) may be to the value of agricultural production at the NUTS2 level. The MC$^3$ algorithm, applied in the study, allowed for easy "capturing" of the models and variables with the greatest explanatory value. The principal function of the algorithm was to randomly sample the regions where the most likely models occur, while ignoring those where the least probable models emerged. Therefore, at this stage, the objective of the study is to present the ranking of the probability models and variables that affect the value of agricultural production at the NUTS2 level in Polish regions.

The main limitation, determining a specific set of factors, is accessibility of statistical data. The database, developed by the authors for the purpose of this study, is derived from a single source (Statistics Poland) in order to ensure data comparability. The only exception are variables that describe the number of regional products. The above presented statistics are provided by the Ministry of Agriculture and Rural Development. The survey takes into account a group of independent variables that represent potential factors responsible for the value of regional agricultural production in 2010–2018.

Bearing the above in mind, the potential factors responsible for the value of agricultural production at the NUTS2 level in the Polish regions can be divided into two groups: (1) one involves variables that describe the condition at the beginning of the research timeframe, (2) another group of factors involves variables presented as averages for the entire period. The application of this group of variables is justified by the requirement of examining the co-variability of the value of agricultural production at the NUTS2 level in Polish regions and other processes, which took place over the analysed time. A list of selected variables is presented in Table 2.

## 3. Results and Discussion

In the course of the research process, the number of iterations in the Monte Carlo simulation equalled 5,000,000, where the first 10% were deemed "burn-in", which means that they served the purpose of eliminating the influence of start-up (initial) values. The number of iterations was considered sufficient, since literature on the subject does not unequivocally determine the number of regression repetitions, and it was empirically evidenced that increasing the number of iterations for the analysed databases did not produce different results.

The total probability of all the models based on the variables presented above is in relation to all the potential models, amounting to 0.42. This means that the models based on the above variables represent 42% of all the possible models. Since this is an unsatisfied value, the authors decided against presenting the models. For the time being, only the probability value of the variable is of importance. The objective of this study was to provide an answer to a question if the number of regional products affects the value of agricultural production at the NUTS2 level in Poland.

It is extremely difficult to capture the cause-and-effect relation in a cross-sectional study, because the results of a quantitative analysis allow us only to determine the co-variability of some processes. Therefore, it is difficult to select the potential factors determining the differentiation of a response variable. The answer to the previously posed question might, therefore, not be very satisfying. Among the variables chosen by the authors, variable X1—number of regional products in NUTS2 regions in 2018—is more probable (see Table 3) than six variables and less probable than three variables. Additionally, variable X2—number of regional products in the NUTS2 regions (av. 2010–2018)—occupies an unsatisfactory position; it is more probable than four variables and less probable than five variables. It does not come as a surprise that the first place in the ranking is taken by the explanatory variable lagged by 8 years. The use of a lagged variable as an explanatory one is common practice in econometrics [44], and it is also justified by the logic of economic phenomena. Similarly, in the case of variable X10—average share of commodity production in agricultural production (av. 2010–2018)—the value of a posteriori probability amounts to 0.701 and is higher than 0.5, i.e., the level for which a variable's significance has been acknowledged. The above presented variables are the only ones at this level, which leads to a conclusion that the value of agricultural production in the NUTS2 regions in Poland is based on "hard" factors, just like the mentioned share of commodity production. Despite large-scale activities promoting regional products, the effort and funds involved in the process, the actual effects of the actions are unsatisfactory. The number of regional products does not affect the output of agricultural production in the region.

**Table 3.** The ranking of variable probability.

| Position in the Ranking | Variable | Posteriori Probability | Averaged Expected Posteriori Value of a Regression Coefficient | Averaged Value of Posteriori Standard Deviation of a Regression Coefficient |
|---|---|---|---|---|
| 1 | X3 | 0.999 | 1.745 | 0.219 |
| 2 | X10 | 0.701 | −97.27 | 79.035 |
| 3 | X8 | 0.209 | 5.456 | 14.332 |
| 4 | X1 | 0.179 | −1.452 | 5.797 |
| 5 | X9 | 0.168 | −0.053 | 0.185 |
| 6 | X2 | 0.144 | 1.214 | 6.973 |
| 7 | X5 | 0.124 | 0.001 | 0.004 |
| 8 | X6 | 0.121 | 20.288 | 923.433 |
| 9 | X7 | 0.099 | −0.017 | 0.93 |
| 10 | X4 | 0.096 | −1.856 | 31.027 |

Source: author's own analysis.

The research in question leads to a conclusion that, in a posteriori probability ranking, variables X1 and X2 are located higher than variable X5—inner investments in biotechnological activity in companies (av. 2010–2018). This means that it is more probable that in a model describing the value of agricultural production in the NUTS2 regions, there are variables related to the number of regional products. Notably, neither of the mentioned variables exceeds a posteriori probability >0.5.

In the literature, there is no clear evidence of how the protected regional products (its number and, therefore, connected marketing activities) contribute to a region's economic and sustainable development. Tourism-related analyses suggest that the consumption of local food by tourists supports the local economy and tourism spending on locally produced goods may stimulate the local economy [6]. The Organisation for Economic Co-operation and Development (OECD, 2012) revealed that food plays an important role in the development of tourism services, since it often comprises 30% or more of tourist expenditure, and this money is regularly spent directly with local business, while a report released by the World Tourism Organisation (2012) stressed the importance of food as immaterial cultural heritage that can enhance the reputation of places worldwide and differentiate locations [45]. To the best knowledge of the authors, none of these studies

and reports, however, analyse the links between introduced GIs schemes and a region's sustainable development that includes an economic perspective and quantitative methods based on a large dataset (as in this case the value of agricultural production). The common conclusions are usually built around the necessity of recognition of ownership and efforts to maintain the link to origin over time and to build up the related reputation—the factors underpinning the right to legal protection of the GI and its promotion [7]. In the authors' opinion, it is valuable to take the step forward and investigate the broader and general perspective of the GI role than only those related to tourism value or development of rural areas. It is important to know if the investment in GI schemes and marketing activities around them will bring general value to the region.

The results of the analysis presented above have some limitations. First and foremost, the small number of regions in Poland determines the number of the dependent variables and, as such, has posed some restrictions to the interpretation of the study. As such, the case of Poland is only the beginning of a broader study of the impact of regional products on the value of agricultural production at the NUTS2 level.

## 4. Conclusions

GIs (and other forms and promotion and protection programmes affecting regional, traditional products) provide a basis for sustainability owing to the relation with the origin and the capacity for "the reproduction of local resources" (FAO, 2010, cited in [7]), i.e., preservation of the territorial, natural and cultural assets that underlie a product's origin-related quality and reputation.

Investing in agricultural production, the development and promotion of food products, needs to be also analysed from an economic perspective. As Vandecandelaere et al. [7] acknowledge, economic development, environmental preservation and social welfare may sometimes be seen as having trade-offs.

Due to a lack of unambiguous, theory-based recommendations, we had to resort to an atheoretical approach. Using the Bayesian approach, the authors have made an attempt to answer a more general question if the number of traditional food products may impact sustainable region development and how marketing should support the process.

In the context of the research, it may be stated that investing in marketing programmes supporting traditional food products may affect the remaining elements of sustainable development more than economy itself (in this case, measured by the size of agricultural production in regions). The specific nature of traditional products, their relations with a location's heritage and the production mode lead to specific limits in the production processes and a finite number thereof. While their features are important elements of sustainable development, their broader economic impact remains insignificant.

The number of traditional food products is limited due to their nature, and one cannot expect the regional agriculture value to be based on traditional regional food products. Probably, only a global brand (of global awareness and scope) would bring about significant economic effects.

This is an indication that in this context, marketing activities should focus primarily on the image aspect, a place brand's link with the nature of the products and showing their importance to the environmental preservation and social welfare of a specific region. Promotion of traditional food products should focus on emphasizing a place's heritage but also their importance to the inhabitants and the future generations. We must not forget the potential of food products in building a unique value to consumers and the important role of food products in creating place brand experiences. Agriculture is evolving from being merely a supplier of raw materials for industrial processing towards a revaluation of gastronomic traditions and the creation of more sophisticated culinary value [46]. Especially, tourists want to see, learn about and experience on-site processes, a desire that can be considered a quest for true authenticity [47], in contrast to the staged authenticity [48] typical of mass tourism. Hence, food products need to be viewed as an element of more

general impressions and opinions, which result in a specific attitude towards a place and specific behaviour (including that leading to strictly economic results).

Investments in the promotional schemes (such as GIs) undertaken by regional or national authorities add foremost emotional value to the consumers and producers alike, which helps the latter to achieve their individual rather than regional economic goals. Their role as suppliers is, however, beyond the economic importance for a region, as a broader, sustainable context is of superior value. As such, traditional food products play a significant role in marketing of a region as a vehicle of communicating a place's features, customs, traditions, identity and part of the region's experience. There is a clear shift from the location of production to a wider and more holistic place and destination branding process [49]. The concept of "foodscapes", which unite local culture, creativity and food is becoming, therefore, relevant in highlighting the important linkages between novelty, authenticity and locality in food experiences [50].

Traditional food may be considered a resource for the local community, because typicity relates not only to the process of production but also to the relations between the different actors in the territorial systems who give the product a collective dimension [51]. The importance of the role and the meaning of traditional food to the region should be, therefore, emphasised among regional stakeholders (farmers, distributors, residents, local politicians, etc.) and reinforced by marketing-related actions. However, this view needs to be supported by further qualitative investigations, as the method chosen is quantitative in nature and does not answer the postulate directly. It would be also valuable to conduct similar research on a European scale to resolve the issue examined in the paper from a methodological and empirical point of view.

**Author Contributions:** Conceptualization, M.F., Literature Review, M.F., J.G., Methodology, J.G., Data Collection, J.G., Software, J.G., Formal Analysis, J.G., Original Draft Preparation, M.F., Review and Editing, M.F., Supervision M.F. All authors have read and agreed to the published version of the manuscript.

**Funding:** This research received no external funding.

**Institutional Review Board Statement:** Not applicable.

**Informed Consent Statement:** Not applicable.

**Data Availability Statement:** Data was collected from Statistics Poland (formerly known in English as the Central Statistical Office; https://stat.gov.pl/en/databases/) and from Ministry of Agriculture and Rural Development in Poland (https://www.gov.pl/web/agriculture).

**Conflicts of Interest:** The authors declare no conflict of interest.

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
