# Peer review of "Traditional Food Products—Between Place Marketing, Economic Importance and Sustainable Development"

_sustainability, doi:10.3390/su13031277_

Round 1

Reviewer 1 Report

Dear Authors,

The manuscript topic is interesting, but the paper structure is very poor. Before publication manuscript should be significantly improved. 

Main remarks:

  1. Structure. The Introduction is very short. The Material and Methods section contains an overview of the literature, not methodology. The Results section contains methods. The Discussion section is not a discussion. The Conclusions section is missing. The layout of the manuscript requires major changes.
  2. Introduction. The introduction is very short, with only 7 references. What is the research gap? What are the purposes of the research?
  3. Materials and Methods.
    1. Most of the content in this section is a literature review (2.2, 2.2, 2.3).
    2. It is difficult to say on what basis the variables were selected. What is the period?
    3. Part of the methodology appears in the results (lines 391-407), why?
    4. Also, some variables are pointers, which I believe is correct. The other variables are numbers - I think they should be converted to pointers. Regions in Poland are varied under land area and population, so results are not comparable.
  4. Results.
    1. The results should be described in more detail. In my opinion, if something is not confirmed, it is also worth presenting (line 409-414).
    2. Why are the relationships between these variables being searched for? It is not clear how the variables were selected. Why are regional products supposed to affect the value of the entire agricultural production? Perhaps it will be easier to find the relationship between regional products and the region's tourism competitiveness.

    3. Line 381:" the objective of the study is to present the ranking of the probability models and variables that affect the value of agricultural production at the NUTS2 level in Polish regions" - This purpose is not directly related to the title of the manuscript
  5. Discussion. The discussion section is not a discussion. The authors discuss only one publication - number 7.
  6. Conclusions. The conclusions section is missing. Where are the implications for politicians, market makers, farmers, consumers, etc.?

Author Response

Dear Anonymous Reviewer,

Thank you for giving us the opportunity to submit a revised version of our manuscript. We appreciate your time and effort to provide feedback on our manuscript and we are grateful for the insightful comments on and valuable improvements to our paper.

In the table attached to this letter, we explain how your remarks were addressed in the revised version of the paper. We responded to each comment individually, indicating how each of them was addressed and amended.

Thank you again for the interest in our research.

Sincerely,
The authors

Reviewer 2 Report

In my opinion, the subject of research presented in the paper is interesting and current. Manuscript is generally well written. After reading the manuscript, three remarks come to my mind for the authors to consider.

  1. The first two subsection of the ‘Materials and methods’ chapter (2.1. The Place Of Origin Effect, Place Marketing and Sustainability and 2.2. Geographical Indicators) fit better with the ‘Introduction’ chapter.
  2. It is worth considering providing the values of the variables included in the research for individual regions. It may be in ‘Supplementary Materials’. This will increase the range of information that can be read at work. For readers interested in these issues, it will be a source of comparisons and use in research.
  3. There are a lot of indirect citations in the manuscript. It would be better to get to the source than to quote someone if it is possible for the authors of the manuscript.

Author Response

(The authors gave the same response as above.)

Reviewer 3 Report

Dear author,

The paper is well organized, but I think that you can check the next recommendations:

  1. The title of the paper does not correspond to the content and it is too long. It should be revised.
  2. Keywords – just include 5-6 key words, exclude the long phrases.
  3. Raw 36 – mistaken word
  4. Raw 38 – 17 UN – please explain the acronym
  5. Introduction section – please include thesis and/or hypothesis; methods; aim; subject; object; structure; limitations just to provide a clear vision for the paper.
  6. Raw 181-182 – there is a wrong symbol
  7. Raw 335 – review the text
  8. Raw 339-355 – the text is good for a textbook, but it should be excluded in a scientific paper, maybe it will be good just to paraphrase/summarize in 2 or 3 sentences.
  9. Raw 361 – include the author’s name of source [39]
  10. R. 428 – X10 – use the same style in Italic
  11. R. 438 – X5 – text in Italic
  12. R. 451 – Conclusion
  13. R. 369 – Discussion and Results

Author Response

(The authors gave the same response as above.)

Round 2

Reviewer 1 Report

Dear Authors,

Thank you for your response to the comments and revised parts.

However, I have a few comments.

  1. The Introduction section is not well organized. Moving a few parts did not solve the problem, there are only 7 references. The part transferred is a literature review (the articles in Sustainability provide a literature review). This part needs to be significantly improved. New references should be added.
  2. The revised Results and Discussion section is still not a discussion (now only publication 44). This part should be significantly improved.
  3. The authors wrote a very similar article (Florek, M.; Gazda, J. Znaczenie produktów tradycyjnych dla regionu - próba oszacowania, Przedsiębiorczość i Zarządzanie, 803 2016, XVII (4), 177-191). Only once does this reference appear in the manuscript. There is no broader discussion or comparison of the results in the manuscript. Has anything changed in the importance of the products for the region?
  4. Regional products are important in tourism. When analyzing the marketing of a place, one should refer to tourism, the benefits of tourism, and the tourism competitiveness of the region. There are many studies on this subject, e.g.:
    1. Bessiere, J.; Tibere, L. Traditional food and tourism: French tourist experience and food heritage in rural spaces. Journal of the Science of Food and Agriculture 2013, 93, 14, 3420-3425.
    2. Cohen, E.; Avieli, N.; Food and tourism. Attraction and ImpedimentAnnals of Tourism Research 2004, 31, 755– 778.
    3. Roman, M.; Roman, M.; Prus, P.; Szczepanek, M. Tourism Competitiveness of Rural Areas: Evidence from a Region in Poland. Agriculture 2020, 10, 569.
    4. Sasu, K.A.; Ephuran, G. An overview of the new trends in rural tourism. Bulletin of the Transilvania University of Braşov Series V: Economic Sciences 2016, 9, 58, 2.
    5. and many others. I propose to write about it in the introduction and/or conclusions.

Author Response

Dear Anonymous Reviewer,

Thank you for the second review. 

In the table, we explain how your remarks were addressed by us and how they were amended in the second revised version of the paper. 
Thank you again for the interest in our research and we hope this version will meet your expectations and our answers will reply to the issues raised. 

In the revised version of manuscript changes are visible in track changes mode (red color in this version). We also double-checked the spelling and several small corrections in this regard. 

Sincerely,
The authors

Round 3

Reviewer 1 Report

Dear Authors,

The revision is satisfactory. 

Best regards.